# Acute Poisoning Readmissions to an Emergency Department of a Tertiary Hospital: Evaluation through an Active Toxicovigilance Program

**DOI:** 10.3390/jcm11154508

**Published:** 2022-08-02

**Authors:** Raúl Muñoz Romo, Alberto M. Borobia Pérez, Rosa Mayayo Alvira, Mikel Urroz, Amelia Rodríguez Mariblanca, Francisco J. Guijarro Eguinoa, Lucia Diaz García, Julio Cobo Mora, Angelica Rivera, Rosario Torres, Antonio J. Carcas Sansuán

**Affiliations:** 1Clinical Pharmacology Department, University Hospital Paz-IdiPAZ, Universidad Autónoma de Madrid, 28049 Madrid, Spain; rmunozromo@gmail.com (R.M.R.); mikel-el@hotmail.com (M.U.); franciscojavier.guijarro@salud.madrid.org (F.J.G.E.); luciadiaz.ucicec@gmail.com (L.D.G.); 2Clinical Toxicology Unit, University Hospital Paz, 28046 Madrid, Spain; rosamayayoalvira@hotmail.com (R.M.A.); ame_arm2@hotmail.com (A.R.M.); mangelicariveran@hotmail.com (A.R.); rmtsantosolmo@gmail.com (R.T.); 3Emergency Department, University Hospital Paz, 28046 Madrid, Spain; julio.cobo@salud.madrid.org

**Keywords:** acute intoxication, hospital readmission, quality of care, toxicological surveillance, clinical toxicology, toxicovigilance

## Abstract

The aim of this study is to investigate hospital readmissions during 1 year after acute poisoning cases (APC), analyze the temporal behavior of early readmissions (ER) (in the month after the index episode) and predict possible ER. A descriptive analysis of the patients with APC assisted between 2011 and 2016 in the Emergency Department of Hospital La Paz is presented, and various methods of inferential statistics were applied and confirmed by Bayesian analysis in order to evaluate factors associated with total and early readmissions. Out of the 4693 cases of APC included, 968 (20.6%) presented, at least one readmission and 476 (10.1%) of them were ER. The mean age of APC with readmission was 41 years (12.7 SD), 78.9% had previous psychiatric pathology and 44.7% had a clinical history of alcohol addiction. Accidental poisoning has been a protective factor for readmission (OR 0.50; 0.26–0.96). Type of toxin (“drug of abuse” OR 8.88; 1.17–67.25), history of addiction (OR 1.93; 1.18–3.10) and psychiatric history (OR 3.30; 2.53–4.30) are risk factors for readmissions during the first year. Women showed three or more readmissions in a year. The results of the study allow for identification of the predictors for the different numbers of readmissions in the year after the index APC, as well as for ERs.

## 1. Introduction

Numerous studies highlight the relevance of hospital readmissions [1,2,3,4,5]. Hospital readmissions provide information on the evolution of patient health status following initial hospital care and can thus be an indicator of the outcome of the care process. The frequency of hospital readmissions has been considered a quality of care index since 1965 that has eased the identification of problems in the care provided [1,2,3,4,5,6,7], the design of multidisciplinary interventions for discharge follow-up and the avoidance of repeated admissions [1,3,4,5,6,7]. Furthermore, hospital readmissions can have a notable economic impact on the health system [1,2,3,4,5,6].

The majority of studies on hospital readmissions have placed their focus on unscheduled readmissions that take place in internal medicine units or clinical services at acute hospitals. The results obtained are mainly from elderly patients and those with comorbidities. These studies conclude that the main causes for readmission in these patients are deterioration in their chronic disease, inadequate outpatient management and adverse treatment events, factors that, in principle, are clearly preventable [1,2,3,5,6,8]. Likewise, it was observed that a significant percentage of readmissions occurred early (within 1 month), especially in those patients who had more than one previous admission due to the aggravation of a chronic problem, inadequate outpatient management, a previous misdiagnosis and/or an adverse effect to the usual treatment [1,2,3,4,5,6].

Acute poisoning cases (APC) are an important health problem [9,10,11,12,13,14,15,16,17,18,19] constituting a notable proportion of emergency hospital admissions. Likewise, it has been observed that patients with APC have a high frequency of readmission [10,11,12,13,14,15,16,17,18,20,21,22,23,24,25]. In order to identify possible predictors of repeat APC, various studies were carried out to analyze admission patterns in patients with APC; nevertheless, 85% of them referred to voluntary poisonings. The results show a recurrence rate of 18% per year. The rate was seen to be independent from the type of APC (autolytic, abusive/recreational), increasing up to 30% in the first year, with a higher incidence in the first month [22].

In 2010, La Paz University Hospital (HULP) implemented an active drug monitoring program called SAT-HULP to record those hospital admissions caused by acute poisoning. SAT-HULP is based on the semi-automated systematic detection of APC treated in the Emergency Department of the General Hospital (ED) [14,16]. In the first five years of operation, 4693 cases of acute poisoning were detected. To examine the acute poisoning readmissions and identify their temporal behavior and associated factors, a study that evaluated the pattern of early readmission (ER, for those occurring in less than 1 month) and late readmissions (LR, readmissions occurring within the first year after an initial episode in the hospital) was conducted.

The following objectives were defined for the study: 


**Main objective**


Estimate the number of readmissions 1 year after an index episode of admission for APC, both detected by the HULP toxicovigilance program during its first five years of operation, as well as their possible predisposing factors.


**Secondary objectives**


Analyze the temporal behavior of ERs as a result of APC in this ED during the period of the study and its predictive capacity.Predict possible ERs through various risk variables, thus obtaining a risk profile for readmission of the patient with APC.

## 2. Methods and Materials

A descriptive cross-sectional and retrospective study with analytical projection was carried out on all APC collected during the first five years of operation of the SAT-HULP program from 1 April 2011 to 1 April 2016. All cases of acute poisoning with readmission episodes during the first year since the index case were selected. Information related to personal history, type of intoxication, circumstances of the intoxication, clinical manifestations and patient destination were recorded for all selected cases. Information on the number of readmissions after the index day and within the study period was also recorded. The types of intoxication defined were suicidal, abusive, accidental and homicidal. Abusive intoxications refer to the use of substances for recreational or addictive purposes, and the accidental type refers to the involuntary nature of intoxication.

A descriptive statistical analysis was performed on the study variables. Quantitative variables were described by the mean (SD) and median (range) and the qualitative variables by the absolute and relative frequencies. For the statistical analysis, the chi-square test was used, or otherwise the Fisher’s exact test. Statistical significance was considered when *p* values were less than 0.05. 

The descriptive statistics were supplemented by devising a time series of the number of monthly ERs. The time series was subsequently transformed by Fourier analysis (from the time domain to the frequency domain) to obtain a periodogram that allowed the presence or absence of seasonal and/or cyclical periods to be identified. To make a short-term prediction, a deterministic self-projective methodology was used to smooth irregularities and fluctuations in the series, thus allowing for a trend analysis. Finally, a curvilinear estimation was performed by saving the predicted values and making future predictions for the fitted ones.

A classification tree tool was used by means of a CHAID (Chi-square Automatic Interaction Detector) algorithm to obtain the profile of a patient with ER. Classification trees are a statistical technique used to explain the variation of a variable with a single response. This is achieved by performing a continuous division of data through a sequential top-down process into homogeneous, exhaustive and mutually exclusive groups. The technique allowed for selection of the variable that separates the most groups under study to grant prediction of the occurrence of an ER based on the variables that the ED commonly works with. 

Subsequently, a survival analysis was performed using Kaplan–Meier curves to estimate the recurrence after the index episode (the first recurrence considered as an outcome), both overall and by type of intoxication, by comparing the survival of the different curves using the Log-Rank test, which takes into account the differences in survival among groups at all points during the follow-up.

A multiple logistic regression technique was employed to identify the predictive factors for repeat APC behavior. The technique served to evaluate the predictive factors existing for the different readmission levels (more than one per year). This approach concerns the case where the response variable is nominal: it can take r values, corresponding to r excluding classes or categories. Furthermore, this approach, as seen for this study, specifically admits a more restrictive view in which such categories can be ordered in an ordinal manner, i.e., from one to more than one per year. The model was adjusted by the backward stepwise method with those variables that had a clear clinical and/or biological significance (and that were stable over the time of the study) using the Likelihood Ratio test to include or exclude variables that contributed significantly to the model. These variables were age, sex, history of addiction, type of addiction, history of psychiatric pathology, type of intoxication, type of intoxicant involved and the association or not with drugs of abuse in the intoxication. Afterward, a Cox regression was applied to simultaneously evaluate the effect of a series of explanatory variables on the first readmission (overall) and/or on the rate of occurrence of the events and the subsequent calculation of the hazard ratio for the risk assessment. 

To address the problem of uncertainty related to the true index episode (if outside the time window) and the possibility of readmissions to another hospital, a Bayesian approach was used to estimate the odds ratio (OR) [26,27,28]. The previous knowledge gained from the literature was considered for the calculation of the most likely interval to contain the OR. [21,22,23,24]. In this method, the researcher is asked to give prior values for both the OR and the 95% probability interval. This interval arises in a certain sense from a “speculation” regarding the ‘’likelihood interval’’. The researcher, who has taken into account various sources of literature to predict the most probable interval to contain the OR, gives the estimation. Assuming that the interval given is only an estimate, we refer to it as a “likelihood interval” and not a “confidence interval”. With that information, an estimate and a later interval are calculated. This method also provides a value to further calculate the probability the later OR has of exceeding it. In this way, Bayes’ theorem would be the bridge to go from an a priori or initial probability regarding a hypothesis to an updated later probability, based on a new observation. 

The data were analyzed using SPSS 22.0 statistical analysis software, except for the Bayesian analysis, which used Epidat 4.2. The study was evaluated by the Ethics Committee of the La Paz University Hospital, Madrid.

## 3. Results

### 3.1. Frequency of Readmissions, Patient Characteristics and Profile of the Patient with ER

During the period from 1 April 2011 (start date of the SAT-HULP program) to 1 April 2016 (5 years later), 4693 APC were identified in the ED through the SAT-HULP system. During that period, 3665021 patients were attended in the ED. Out of the 4693 APC identified, 968 (20.6%) had at least one readmission, 476 of them being ER (10.1% of the total). The average number of total ERs and LRs is 1 (0-28, IQR: 28). A full description of the characteristics of the patients who presented at least one readmission and the profile of the intoxications is summarized in Table 1. The mean age of the APC with readmission was 41 years (12.7 SD), with a slight predominance of females (51.1%). A significant frequency (78.9%) of subjects with previous psychiatric pathology was observed; as well, 44.7% of the patients presented a history of alcohol addiction. Eighty-six percent of the total number of patients presented symptoms on readmission. The most frequent clinical manifestations were neurological (82.8%), while 43.5% of the total presented behavioral disorders.

The time series produced (Figure 1) showed that the number of ERs per month ranged from 9 to 38 in total. The obtained periodogram showed the absence of clear seasonality, which led to application of a Holt linear smoothing model, which predicted the behavior of the number of ERs for April 2016–April 2018. When analyzing the trend, the model that best fitted was the cubic model, with an R^2^ of 0.877 and with a high significance of the values (*p* < 0.001), which indicates that the ER will present a progressive rise and subsequent decline, remaining stable in the medium term. The fitting equation is: Z(t) = 10.050 + 1.086t − 0.026t^2^, and it is shown in Figure 1.

2.The application of the CHAID algorithm allowed us to observe the conclusions of the displayed tree (available as Appendix A), which are:The variable “history of addiction” is the best predictor for ER.The highest probability of ER (22.8%) is found among those with a history of alcoholism or opiate addiction, a psychiatric history (26.7%), or those that take drugs of abuse as the type of drug involved (29.9%).The lowest probability of ER (6.1%) is among those with no addiction or psychiatric history (3.1%). If these patients have a psychiatric history or are unaware of it, the probability of ER rises to 7.7%.We emphasize that patients who present multiple addictions as a background have an ER probability of 17.2%, while those who present an association of medications and drugs of abuse as intoxicants involved have a 23.1% probability of ER.

Thus, the model correctly classifies 89.9% of the individuals in general with a standard error of 0.004 in the risk assessment, which we consider statistically adequate.

### 3.2. Predisposing Factors for LRs (Total) and ERs

Figure 2 shows the Kaplan–Meier curves with the cumulative survival of ER and LR, after the index episode, for each type of APC.

The average time without a first readmission was 21 months (standard error: 1.06; range: 18.92–23.08). A first readmission was observed at the end of the first year for 121 of the 4693 patients in total. The results of the Long-Rank test for the different types of intoxication showed a chi-square of 12.41 with a clearly significant p (*p* = 0.002), indicating heterogenicity for the “suicidal” type (82.4%) and the “abusive/recreational” type (89.3%), types presenting the highest number of events (first readmission after the index admission).

Table 2 shows the results of the multiple logistic regression where the type of “accidental” intoxication is observed as a protective factor to avoid at least one readmission in a year with an OR of 0.50. Not having a history of addiction has also been seen as a protective factor to avoid two readmissions per year with an OR of 0.32. Conversely, the type of intoxicant “drugs of abuse” and the psychiatric history ‘’unknown’’ and ‘’present’’ are shown as risk factors for readmission in one year, with an OR of 8.88, 1.64 and 3.30, respectively. Furthermore, the history of addiction “multiple addictions” and “alcohol” has also been identified as one of the risk factors for readmission in a year, with an OR of 1.93 and 1.60, respectively. In addition, psychiatric history “unknown” and “present” are shown as risk factors for having two readmissions in one year, with an OR of 3.77 and 7.13, respectively, as well as having a history of “alcohol” addiction, with an OR of 1.96. Likewise, not having drugs of abuse associated with the toxins involved meant an OR of 4.91 for two readmissions in one year. Finally, the following have been shown as risk factors for three or more readmissions in a year, “female” sex with an OR of 1.38, psychiatric history “unknown” and “present” with an OR of 5.66 and 10.88, respectively, history of “multiple addictions” and “alcohol” with an OR of 2.68 and 4.59, respectively, and an OR of 9.01 for those who were not associated with drugs of abuse were shown as risk factors for three or more readmissions in a year.

Cox regression results showed that both the type of intoxication “suicidal” and the type of intoxicant that involved “drugs of abuse” are robustly associated with the risk of readmission during the first year. In the case of patients with intoxications of “suicidal” type, the risk is multiplied by 1.42 times, while for those who present the type of intoxication “drugs of abuse”, the risk is increased by a factor of 9.81, as shown in Table 3. It is worth noting the effect of interactions between history of alcohol addiction and unknown psychiatric history, history of opiate addiction and unknown psychiatric history, history of multiple addictions, alcohol addiction, opiate addiction, cocaine and cannabis addiction with the presence of psychiatric history. The HZ for the different associations multiply the risk by a factor of 4.24, 10.05, 2.72, 5.72, 7.89, 4.89 and 4.64, respectively. There were no differences when adjusting for sex. Likewise, the interaction of the absence of a history of addiction with the presence of psychiatric conditions was significant with an HR of 2.37.

### 3.3. Bayesian Analysis Results

As mentioned in the methods and materials section, in the Bayesian analysis (Table 4) we assume an OR value and a confidence interval established a priori based on previous studies: OR = 1.17 (0.78, 1.76) (psychiatric history) and OR = 2.06 (1.07, 3.97) (drug abuse intoxication), respectively [21,22,23,24].

This analysis was carried out specifically for psychiatric history ‘’present’’ and type of toxin “drugs of abuse” as risk factors for having a readmission at one year, as referenced in the literature. As it can be seen, the probability that the OR in the first case is less than 2.00 is zero, while the probability that it does not reach 3.5 is 84.1%. For the second case, the probability of it being less than 1.10 is only 24.5%, while the probability of it being less than 9.00 is 100% (our OR is close to 9). Thus, the analysis yields OR values of 8.88 and 3.30 for the first and second cases, respectively. According to this approach, the presence of a psychiatric history would multiply by 3.5 the possibilities of having at least one readmission in the first year and that of presenting “drugs of abuse” as an implicated toxic would do so by 9. These results confirm the previously shown data obtained in the inferential analysis.

## 4. Discussion

At the overall descriptive level, the main conclusion of our study is that the profile of the acutely intoxicated patient requiring readmission is a woman between thirty and fifty years of age, who is attended mainly in the hospital emergency department for less than 24 h, with little need for admission to other units. Intoxication is most frequently seen from alcohol, benzodiazepines and cocaine.

At an analytical level, we considered the possibility of analyzing the pattern of repeated APC detected through a hospital-based active drug monitoring program. In addition, knowledge of the frequency of self-harm attempts helped us to understand the problem (trigger of most APCs) by identifying possible predictors of recurrence of such episodes. We observed that for all patients, the predictors of readmission were intoxication by drugs of abuse, history of psychiatric pathology and previous addiction to alcohol, in addition to multiple addictions for two readmissions in the same year. These results are consistent with those found in the international literature [8,18,20,21,22,23,24,25]; moreover, our group already obtained similar results when analyzing readmissions for APC using DRGs (Diagnosis Related Groups) [16]. Here, we were able to observe that patients who were readmitted suffered a less severe intoxication and presented less severe medical comorbidities, both in the index episode and upon readmission. Although these patients have a more favorable clinical pathway, they are at higher risk of presenting psychiatric comorbidity and more severe suicide attempts (DRGs 750, 428 and 426) [8,16,17,18,20,21,22,23,24,25]. 

The absence of seasonality in readmissions and the linear trend detected made it possible to observe the predictable behavior of ERs and monitor the possible appearance of anomalous increases. We believe that this first temporal approximation to the phenomenon of ER in APC is interesting, but this analysis will be more robust when we are able to gather data for 6 to 10 full years, as this volume will allow us to apply models that use a stochastic and non-deterministic point of view [29]. It is also relevant to highlight the use of a Bayesian approach to deal with the uncertainty inherent in the nature of the problem, as described in the Methods and Material section. This meant that the information provided by previous studies in the same direction could be included in the analysis [22,23,24]. The results thus obtained were in agreement with those of the multivariate analysis, which means that we can confirm the consistency of the data obtained in the present study.

At this point, it is necessary to consider as one of the main limitations of our study the fact that it is a single institute and regional study; therefore, it may be subject to bias. However, we think that our center is quite representative of the general picture in our country. Its reference area population is about 752,006 inhabitants, and although it serves a mostly urban population, it also gives medical coverage to the rural population in the north of the Autonomous Community of Madrid. Furthermore, our data are consistent with epidemiological studies conducted in our environment as referred in the literature [9,10,11,12,13,14,16,17,18]. Another limitation to consider would be the cross-sectional methodological nature of the study, in which data collection refers to a specific moment in time, without an induction and latency period between exposure and outcome being assumed. Despite this, cross-sectional studies can be of great value when studying risk factors that do not change over time, because we are more certain that the time sequence is better respected. In this study, this would be the case observed for gender and a history of mental illness and addiction. We understand that by their nature, the latter remains fairly stable over time.

Gender was not a significant predictor for ER or LR, even when adjusting for other variables in the study, but it was a significant predictor for having three or more readmissions during the same year. This confirms the findings of many studies of parasuicidal APC and is consistent with others arguing that women tend to repeat self-harm attempts more often than men [10,11,16,18,21,22,23,24]. Age was not a predictor of readmission during the first year in our study, although many results in the literature show the highest rate of recurrence in the age range of 25–54 [8,9,11,12,18,21,22,23,24,25]. In the existing literature, we observed that age, sex, area of residence and type of physician were discarded in the initial care. In our study, it was also detected that a previous episode to attempt self-harm would constitute a risk factor for a new attempt, while the fact of having several previous suicide attempts would be a greater risk factor than an isolated episode, which is similar to what has been described in the literature [8,16,21,22,23,24,25].

Non-association with drugs of abuse in APC is a robust predictor for having two, three or more readmissions in a year. This may be explained by a possible relationship between two competing risks: mortality and readmission. The patients who are most often readmitted are those who are addicted to alcohol and other drugs of abuse, where these are not associated with prescribed medications or other potentially lethal products, as in the case of suicidal intents (also reflected in the literature) [8,12,16,18]. This paradoxical effect may be even stronger in short-term outcomes. 

Likewise, we were able to identify various predictors and their interactions for the different levels of readmission per year (2, 3 and more than 3 in a year), and to classify and obtain a profile of patients with ER based on the variables presented on admission to the ED. The latter data are consistent with reports in the literature that also identify these predictors for drug-induced APC; however, there is some disparity for benzodiazepine-induced APC. The positive association found with the variables identified in our study was not found in some of the different sources consulted [23,24,25]. This latter observation, which is predictable given the variability and complexity of the study phenomenon, does not prevent us from maintaining that the information obtained on the type of drug used in APC helps to identify patients at risk of future events.

Our study provides valuable information for implementing a clinical management strategy for intoxicated patients, managed under the “Triple Aim” model [30], which facilitates the segmentation of the population and the use of predictive models that identify their different needs as defined by this strategy [30,31]. Thereby, needs can be anticipated proactively rather than reactively, through the use of predictive models that are necessary for the redesign of the services involved [31,32]. In this regard, we have already calculated the cost impact of APC on the Spanish National Health System (SNHS). We were able to show that a considerable figure within the Total Consolidated Public Health Spending of the SNHS for the corresponding year goes to APC [16]. The population between 15 and 45 years of age accounts for 50% of these costs. This study confirms this age-related data, and it is therefore here where most efforts in preventive strategies and health services should be focused [16,18,33,34].

It is also necessary to consider, as one of the main actions to manage the problem of readmission in patients with APC, that these patients are therefore candidates for assessment and subsequent treatment by the psychiatry unit, both in the acute phase and the medium to long- term, as seen before. By observing the costs per survivor without readmission, we see that these patients require a significant amount of resources from the system [16]. As a result, the Department of Psychiatry plays a special role in the process of comprehensive care for APCs, thereby highlighting one the strengths of our center’s organizational care: All APC that present autolytic attempts are evaluated psychiatrically. This policy supports the need for the multidisciplinary nature of the CTU, as it has been shown in recent studies.

In light of these results, we observe that active toxicovigilance programs are important tools for the in-depth analysis of the evolution of APC in our environment and the resulting consumption of resources for the health system [14,16]. It is, therefore, a useful tool for the management of the quality of the services involved in the care of intoxicated patients, thus allowing detailed analyses of readmissions as a relevant indicator of health outcomes in these patients.

## 5. Conclusions

The profile of APC with readmission is a woman between 30 and 50 years of age who barely needs admission to units other than the ED. The most commonly used toxic substances are alcohol, benzodiazepines and cocaine.The results of the study allow us to identify predictors for different numbers of readmissions in the year after the index APC as well as for ERs; the detailed analysis of readmissions is thus revealed as a critical indicator of health outcomes in poisoned patients.The SAT-HULP thus makes it possible to assess, in depth, the evolution of APC in their environment. Active toxicovigilance programs are important tools for in-depth analysis of the evolution of APC, the consumption of resources, and for evaluation and improvement of the quality of the services involved in the care of intoxicated patients.

## Figures and Tables

**Figure 1 jcm-11-04508-f001:**
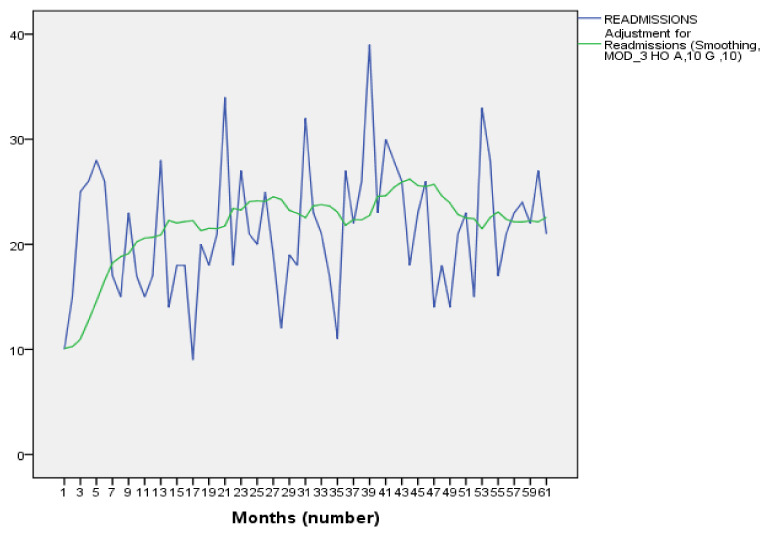
Time series of monthly ERs.

**Figure 2 jcm-11-04508-f002:**
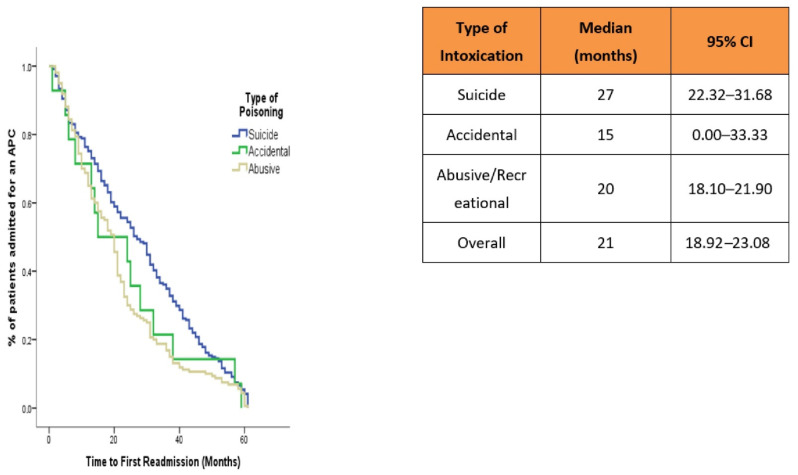
Time elapsed from admission to the ED for APC to readmission. Survival analysis using Kaplan–Meier curves.

**Table 1 jcm-11-04508-t001:** Patient characteristics by type of intoxication ^1^.

	Suicide (*n* = 474)	Abusive (*n* = 476)	Accidental (*n* = 18)	Grand Total (*n* = 968)
**Sex (*n* (%) women)**	300 (60.6)	187 (37.7)	8 (1.6)	495 (51.1)
**Age (mean (SD))**	39.6 (14.7)	41.3 (15.3)	69.1 (23.3)	41.0 (17.8)
**History of addiction (*n* (%) YES)**	291 (43.7)	373 (43.0)	2 (0.3)	666 (68.8)
**Psychiatric history (*n* (%) YES)**	448 (58.6)	313 (40.9)	3 (0.4)	764 (78.9)
**Type of intoxicant (*n* (%))**				
**Medication**	378 (91.0)	22 (5.3)	15 (3.6)	415 (42.9)
**Drug of abuse**	92 (16.8)	454 (82.9)	1 (0.2)	547 (56.5)
**Domestic product**	4 (80.0)	-	1 (20.0)	5 (0.5)
**Industrial product**	-	-	-	-
**Other (includes food poisoning)**	-	-	1 (100.0)	1 (0.1)
**Symptoms on admission (*n* (%) YES)**	359 (43.1)	466 (56.0)	7 (0.8)	832 (85.9)
**Analytical determination (*n* (%) YES)**	146 (62.4)	72 (30.8)	16 (6.8)	234 (24.1)
**Digestive decontamination (*n* (%) YES)**	204 (94.4)	11 (5.1)	1 (0.5)	216 (22.3)
**Use of antidote (*n* (%) YES)**	100 (81.3)	16 (13.0)	7 (5.7)	123 (12.7)
**Patient destination (n (%))**				
**Discharge from ED**	342 (43.1)	434 (54.7)	17 (2.1)	793 (81.9)
**Admission to ICU**	10 (90.9)	1 (9.1)	-	11 (1.1)
**Admission to the Medicine Department**	5 (38.5)	8 (61.5)	-	13 (1.3)
**Death**	-	-	-	-
**Transfer**	46 (83.6)	8 (14.5)	1 (1.8)	55 (5.7)
**Voluntary Discharge**	10 (32.5)	21 (67.7)	-	31 (3.2)
**Admission to the Psychiatric Unit**	28 (90.3)	3 (9.7)	-	31 (3.2)

^1^ No homicide cases were found. ED: emergency department; ICU: intensive care unit; SD: standard deviation. The differences found are significant with a *p* value < 0.05. The percentage of cases that present the variable for each type of intoxication was calculated on the total of each variable (last column) and that of the latter on the total of all cases.

**Table 2 jcm-11-04508-t002:** Multivariate analysis. Results of multiple logistic regression.

	OR	CI 95%	*p* Value
**At least 1 Re-admission per year (*n* = 516)**			
Psychiatric History			
Unknown *	1.64	1.05–2.56	0.030
Yes	3.30	2.53–4.30	<0.001
No (Ref)			
Addiction Background			
Multiple addiction	1.93	1.18–3.16	0.009
Alcohol	1.60	1.04–2.46	0.031
Tobacco (Ref)			
Type of Intoxication			
Accidental	0.50	0.26–0.96	0.036
Abusive/Recreational (Ref)			
Type of Intoxicant			
Drugs of Abuse	8.88	1.17–67.25	0.035
Other (Ref)			
**2 readmissions per year (*n* = 185)**			
Psychiatric History			
Unknown	3.77	1.81–7.83	<0.001
Yes	7.13	4.19–12.13	<0.001
No (Ref)			
Addiction Background			
No	0.32	0.17–0.62	0.001
Alcohol	1.96	1.03–3.72	0.040
Tobacco (Ref)			
Intoxicant-Drugs of Abuse Combination			
No	4.90	1.72–13.95	0.003
Yes (Ref)			
**3 or more readmissions per year**			
Sex			
Female	1.38	1.03–1.84	0.031
Male (Ref)			
Psychiatric History			
Unknown	5.66	2.94–10.87	<0.001
Yes	10.88	6.49–18.21	<0.001
No (Ref)			
Addiction History			
Multiple addiction	2.68	1.14–6.30	0.023
Alcohol	4.59	2.15–9.81	<0.001
Tobacco (Ref)			
Combination of Intoxicants-Drugs of Abuse			
No	4.90	1.72–13.95	0.003
Yes (Ref)			
**3 or more readmissions per year (*n* = 267)**			
Sex			
Female	1.38	1.03–1.84	0.031
Male (Ref)			
Psychiatric History			
Unknown	0.33	2.94–10.87	0.000
Yes	0.26	6.49–18.21	0.000
No (Ref)			
Addiction History			
Multiple addiction	2.68	1.14–6.30	0.023
Tobacco (Ref)			
Intoxicant-Drugs of Abuse Combination			
No	9.01	2.76–29.38	0.000
Yes (Ref)			

* The presence or absence of a psychiatric history has not been recorded.

**Table 3 jcm-11-04508-t003:** Results from the Cox regression analysis of the first readmission within one year after the index episode (*n* = 414).

	HR	CI 95%	*p* Value
**Type of Intoxicant Drugs of Abuse**	9.81	1.29–74.74	0.028
**Type of Suicidal Intoxication**	1.42	1.02–1.96	0.035
**AH Alcohol * PH Unknown Interaction**	4.24	1.85–9.75	0.001
**AH Opioid * PH Unknown Interaction**	10.05	1.39–72.61	0.022
**AH Multiple addiction * PH Present Interaction**	2.72	1.87–3.96	<0.001
**AH Absent * PH Present Interaction**	2.37	1.76–3.19	<0.001
**AH Alcohol * PH Present Interaction**	5.72	4.27–7.68	<0.001
**AH Opioid * PH Present Interaction**	7.89	3.65–17.08	<0.001
**AH Cocaine * PH Present Interaction**	4.89	2.87–8.34	<0.001
**AH Cannabis * PH Present Interaction**	4.64	2.33–9.23	<0.001

* means interactions between risk factors. AH: addiction history; PH: psychiatric history; HR: hazard ratio.

**Table 4 jcm-11-04508-t004:** Results of the Bayesian analysis of all patients.

Odds Ratio Estimation				
**Data: A priori probability interval (95%)**	**1st Case**		**2nd Case**	
**OR**	1.170		2.060	
**Lower Limit**	0.780		1.070	
**Upper Limit**	1.760		3.970	
**Data: Empirical confidence interval (95%)**				
**OR**	8.880		3.300	
**Lower Limit**	1.170		2.415	
**Upper Limit**	67.250		3.949	
**Probability (OR > X) for selected points**				
**Point**	1.10	9.00	2.00	3.50
**Probability**	0.755	0.000	1.000	0.159

## Data Availability

Individual participant data will be made available on request to the corresponding authors. After approval of a proposal, data will be shared through a secure online platform.

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
