# Peer review of "Acute Poisoning Readmissions to an Emergency Department of a Tertiary Hospital: Evaluation through an Active Toxicovigilance Program"

_jcm, 2022, doi:10.3390/jcm11154508_

Round 1
Reviewer 1 Report
Dear Editor,
Thank you for inviting me to review this article again. Despite the authors' efforts to improve the manuscript, I do not see what this single-center study adds to the already published studies.
Author Response
As we recognized in the previous response to the reviewer, the limitations due to the unicentric nature of the study are acknowledged in the discussion section. In any case, we still think that our centre is quite representative of the general picture in our country.
Reviewer 2 Report
The manuscript in its present form has improved.
Author Response
Thanks a lot for your comment.
This manuscript is a resubmission of an earlier submission. The following is a list of the peer review reports and author responses from that submission.
Round 1
Reviewer 1 Report
Dear Authors,
I read with interest your article "Acute Poisoning Readmissions to an Emergency Department of a Tertiary Hospital: Evaluation through an Active Toxicovigilance Program".
Major concerns:
- This study is single-center and does not reflect the full range of care that may be accessed by "readmitted" patients, although the authors have attempted to correct for this bias through statistical methods. The conclusions of the study for such an issue are very limited by the sample size.
- English language needs major revision.
Other concerns
- Introduction:
* several errors in the typography of references. The square brackets are missing several times.
* Page 1 line 29: One or more references should be put after "readmissions
* Page 2 line 52: The sentence at the end of line 52 is not clearly understandable.
Page 2 line 56: This last paragraph seems to me to be more methodical than in the introduction.
- Method:
* This paragraph should include the description of your database. At the end of the article it says that patient consent was not required; were patients informed of the collection and possible use of their data?
- Results :
* Page 3 line 147: You talk about average but you put in parenthesis IQR? What does this mean? Per patient?
* Table 1: The table is not very well presented. It is not always clear what the denominator of the percentage is.
* In your introduction you criticize that the majority of the studies carried out on the subject refer essentially to voluntary intoxications ... more than 95% of the reported intoxications are voluntary?
* Figure 2: you indicate in table 1 that there are no homicides and I find some in figure 2?
Discussion:
* You should discuss the limits of your article ...
Reviewer 2 Report
Authors presented a study evaluating predictive factors associated with repeated acute poisoning and readmissions to Emergency.
There are some comments:
- Methods Section should be improved by: 1) Explaining more clearly the criteria used to define an APC (clinical?); 2) Where clinical data was obtained and who registered; there were agreement between researchers?; 3) Define in the text the 3 types of intoxication, mostly the difference between abusive and accidental.
- In the Results Section, the total number of attended emergencies during the study period would be interesting to be presented.
- Table 1: Are there statistically significant differences between the characteristics among 3 types of intoxications?
- Abbreviatures of tables should be explained at the bottom of the tables.
- Some reccommendations to apply to avoid repeated APC should be explained in the Discussion Section.
- The number of the reference in the text are presented in different styles and without a numerical order.